# Current Status of Cell-Based Therapy in Patients with Critical Limb Ischemia

**DOI:** 10.3390/ijms21238999

**Published:** 2020-11-26

**Authors:** Frantisek Jaluvka, Peter Ihnat, Juraj Madaric, Adela Vrtkova, Jaroslav Janosek, Vaclav Prochazka

**Affiliations:** 1Department of Surgery, University Hospital Ostrava, 708 52 Ostrava, Czech Republic; frantisek.jaluvka@fno.cz (F.J.); peter.ihnat@fno.cz (P.I.); 2Department of Surgical Studies, Faculty of Medicine, University of Ostrava, 703 00 Ostrava, Czech Republic; 3Clinic of Angiology, Faculty of Medicine, Comenius University in Bratislava, 813 72 Bratislava, Slovakia; madaricjuraj@gmail.com; 4Department of Applied Mathematics, Faculty of Electrical Engineering and Computer Science, VSB—Technical University of Ostrava, 708 00 Ostrava, Czech Republic; adela.vrtkova@vsb.cz; 5Faculty of Medicine, University of Ostrava, 703 00 Ostrava, Czech Republic; janosek@correcta.cz; 6Radiodiagnostic Institute, University Hospital Ostrava, 708 52 Ostrava, Czech Republic; 7Institute of Imaging Methods, Faculty of Medicine, University of Ostrava, 703 00 Ostrava, Czech Republic

**Keywords:** cell therapy, peripheral arterial disease, bone marrow, adipose tissue, mesenchymal stem cells, exosome, critical limb ischemia

## Abstract

(1) Background: The treatment of peripheral arterial disease (PAD) is focused on improving perfusion and oxygenation in the affected limb. Standard revascularization methods include bypass surgery, endovascular interventional procedures, or hybrid revascularization. Cell-based therapy can be an alternative strategy for patients with no-option critical limb ischemia who are not eligible for endovascular or surgical procedures. (2) Aims: The aim of this narrative review was to provide an up-to-date critical overview of the knowledge and evidence-based medicine data on the position of cell therapy in the treatment of PAD. The current evidence on the cell-based therapy is summarized and future perspectives outlined, emphasizing the potential of exosomal cell-free approaches in patients with critical limb ischemia. (3) Methods: Cochrane and PubMed databases were searched for keywords “critical limb ischemia and cell therapy”. In total, 589 papers were identified, 11 of which were reviews and 11 were meta-analyses. These were used as the primary source of information, using cross-referencing for identification of additional papers. (4) Results: Meta-analyses focusing on cell therapy in PAD treatment confirm significantly greater odds of limb salvage in the first year after the cell therapy administration. Reported odds ratio estimates of preventing amputation being mostly in the region 1.6–3, although with a prolonged observation period, it seems that the odds ratio can grow even further. The odds of wound healing were at least two times higher when compared with the standard conservative therapy. Secondary endpoints of the available meta-analyses are also included in this review. Improvement of perfusion and oxygenation parameters in the affected limb, pain regression, and claudication interval prolongation are discussed. (5) Conclusions: The available evidence-based medicine data show that this technique is safe, associated with minimum complications or adverse events, and effective.

## 1. Introduction

Most regulatory authorities have realized the need for early access to innovative therapies for unmet medical needs that may improve health care quality for life-threatening conditions. Since critical limb ischemia is a significant healthcare problem with its rising incidence, accelerating the development of innovative advanced therapies is essential. However, cell therapy’s therapeutic efficacy in various animal models has been only partially reproduced in human clinical trials. Despite the progress in basic and clinical research over the last two decades, the regenerative therapy of limb ischemia based on cell therapy is still considered an experimental treatment method, not recommended for routine clinical use.

### 1.1. Critical Limb Ischemia

Peripheral arterial disease is a severe medical condition associated with high morbidity, mortality, and significant socioeconomic and social impacts. Increased incidence of renal failure, cardiovascular disease, diabetes mellitus with associated microangiopathy and retinopathy are documented in patients with PAD [1,2].

Critical limb ischemia (CLI) is the end-stage of peripheral artery disease (PAD) caused by tissue hypoxia and characterized by ischemic rest pain, ulcers, or gangrene associated with a significant risk of affected limb loss and a high risk for cardiovascular events. The annual incidence is approximately 500–1000 new cases per million in industrialized countries. The disease prevalence increases with increasing rates of diabetes, aging of the population, and persistent rates of tobacco abuse [3]. Current treatment options are based on endovascular intervention, bypass surgery, and the best medical conservative treatment with infusion of vasoactive agents as a potential adjunct therapy [4,5]. However, approximately 20–30% of critical limb ischemia patients are not eligible for revascularization, or this procedure has failed. Besides, the mortality of patients with CLI is about 20% within 6 months from diagnosis, the 1-year mortality of patients with no-option CLI climbs to 40%, reflecting the multivascular character of atherosclerosis in this population [6,7].

Over the past two decades, many therapeutic advances have been accomplished in the field of PAD revascularization techniques. Modern technologies improve short-term outcomes of interventions, though they fail to improve the long-term expectations. Moreover, although surgical or endovascular revascularization improves macrovascular perfusion, microvascular perfusion often remains unimproved. Supportive treatment is also being used to treat patients with PAD, usually with hyperbaroxia and lumbar sympathectomy. Despite all advancements in medicine, literature sources show that 20–45% of patients are not suitable for a revascularization procedure [8,9]. This subgroup of patients is burdened with a high risk of limb loss, increased morbidity, and mortality.

Thus, there is a critical need to develop novel therapeutic strategies to improve limb perfusion and healing process, mainly for patients without revascularization options. New methods targeting neovascularization and microcirculation improvement provide a potential solution and new hope for no-option CLI patients [10].

### 1.2. Cell Therapy

The concept of cellular therapy has made significant progress over the past decades, from the use of cells for its original function (red blood cell transfusion) toward their use for treatments different from their native role. In 1997, Asahara et al. identified a class of bone marrow (BM) progenitors responsible for angiogenesis in ischemic tissues [11]. The first test using autologous mononuclear bone marrow cells in patients with CLI was published by Tateishi-Yuyama et al. in 2002 [12]. The mechanism of cell therapy action consists of accelerating the body’s regeneration processes in ischemic tissues. The developmental and regenerative processes of the vascular system are divided into two steps. The first one is the process of vasculogenesis, i.e., an in-situ differentiation of cellular precursors of angioblasts into the form of primitive endothelial cells, which subsequently form a labyrinth of primitive vessels (see Figure 1). The second step is angiogenesis, which includes the growth and remodeling of the primitive network into a complex one. Therapeutic angiogenesis is applied particularly in adulthood [13].

Multifunctional stem cells are nonmature, nondifferentiated tissue precursors. Their essential characteristics are the high potential for self-renewal and their ability to differentiate into various cell lines based on stimuli from the surrounding environment. Stem cells are present in almost all types of tissues and represent crucial components of endogenous repair mechanisms. Adult stem cells include bone marrow stem cells, circulating stem cells, and tissue-resident stem cells. Bone marrow-derived stem cells contain a heterogeneous group of hematopoietic stem cells, giving rise to all cell lines of the hematopoietic system, multipotent adult progenitor, mesenchymal or stromal cells. They are crucial for tissue regeneration with important paracrine activity, and, finally, side population cells [14].

Numerous stem cell populations from various sources have been proposed for cell-based therapy; however, endothelial progenitor cells (EPCs), mesenchymal stem cells (MSCs), and their products play a pivotal role in therapeutic neovascularization and treatment of limb ischemia.

The use of cell therapy in treatment of critical limb ischemia is currently undergoing tumultuous development and although several reviews on this issue have been published, many of those are outdated by now. Additionally, several meta-analyses have been performed; none of the reviews, however, focused specifically on these meta-analyses and their detailed comparison. In this narrative review, we collected the current information and aimed to provide an up-to-date critical overview of the knowledge and evidence-based medicine data on the position of cell therapy in the treatment of PAD. This review should serve as a source of information both to the professionals only entering this field and experienced expert looking for concise up-to-date information.

## 2. Materials and Methods

Literature sources for this narrative review were identified through a search in the Cochrane and PubMed databases in September 2020. The keywords “critical limb ischemia cell therapy” were used for the initial search. Altogether, 589 papers were identified, 11 of which were reviews and 11 meta-analyses (see Table 1). The meta-analyses and reviews were used as the primary source of information for this paper, using cross-referencing for identification of additional papers and validation of the information to be included in this review. In all, 59 papers were used for this review.

## 3. Cell Types and Methods Used in the Cell Therapy of CLI Patients

### 3.1. Endothelial Progenitor Cells (EPCs)

EPCs are present in the BM as a part of the mononuclear hematopoietic cell fraction; they also circulate in peripheral blood and are found in the form of resident stem cells in almost all tissues capable of differentiation within the endothelial lineage. Endothelial and hematopoietic lineages have common precursors, hemangioblasts. Under physiological conditions, the number of circulating EPCs is small. Their number in the peripheral blood increases in response to ischemia by mobilization from the BM after secretion of proangiogenic cytokines, such as the vascular endothelial growth factor (VEGF), stromal cell-derived factor 1 (SDF-1), or hypoxia-inducible factor 1 (HIF-1). EPCs are attracted to ischemia sites and contribute to angiogenesis by secreting interleukins, growth factors, and other cytokines by activating resident stem cells, recruiting circulating progenitors, and inhibiting cells apoptosis. Altogether, through these indirect mechanisms, the therapeutic cells accelerate the formation of the vascular network and enhance healing processes [25].

Despite the progress in stem cell research, the precise definition of EPCs remains uncertain and controversial. In general, they are characterized by the coexpression of markers for both hematopoietic and endothelial cell lineages (CD34, CD133, VEGF receptor-2, kinase-insert domain receptor, von Willebrand factor, and endothelial nitric oxide synthase) [26]. CD133 is an early hematopoietic stem cell marker. The loss of CD133 expression is associated with increased expression of a variety of endothelial lineage markers constituting a signal for EPCs maturation toward the endothelial lineage [27]. Expression of such markers can distinguish between early EPCs (e.g., CD133 + CD34+ cells) and late-outgrowth EPCs. While the subpopulation of late-outgrowth EPCs form vascular networks de novo and can incorporate into nascent blood vessels, early EPCs indirectly augment vasculogenesis via the paracrine mechanism [25,26,27,28].

The number of administered CD34+ cells has been shown to influence cellular therapy’s clinical benefit [29,30]. Klepanec et al. showed that the number of administrated CD34+ cells, unlike the total number of bone marrow mononuclear cells (BM-MNCs), was strongly related to clinical benefit [30]. This result partially contradicts the findings of the (PROVASA) study—Intraarterial Progenitor Cell Transplantation of Bone Marrow Mononuclear Cells for Induction of Neovascularization in Patients with Peripheral Arterial Occlusive Disease trial. The number of CD34+ cells and BM-MNCs were shown to be independent predictors of improved ulcer healing [31].

Conversely, results published by Klepanec et al. [30] are consistent with studies demonstrating the superior effects of enriched CD34+ cells compared with BM-MNCs. The CD34+ cells were able to restore microcirculation and improve tissue perfusion in preclinical models, [32] as well as in clinical series [29,33]. In fundamental studies, the surface expression of CD34+, CD133+, and VEGF receptor-2 and kinase insert domain receptor identified a population of EPCs with enhanced potency for neovascularization of ischemic tissue [32,34]. Furthermore, enriched CD133+ progenitor cells demonstrated positive functional effects in patients with chronic as well as recent myocardial infarction [35,36]. The notion that mononuclear cells depleted of CD34+ cells do not improve myocardial function in a murine infarct model further supports the hypothesis that CD34+ cells may be pivotal for therapeutic benefits [37].

### 3.2. Mesenchymal Stem Cells (MSCs)

MSCs are nonhematopoietic cells present in the bone marrow, adipose tissue, and many other tissue sources. MSCs, as stromal cells, constitute an essential part of the marrow microenvironment supporting hematopoiesis, also possessing extensive proliferative capacity [38]. They have multilineage potential with the ability to differentiate into adipogenic, osteogenic, chondrogenic, and skeletal muscle cells, as well as into vascular smooth muscle cells, neural precursors, cardiomyocytes, or perivascular cells [39,40]. It is now well accepted that mesenchymal stem cells (MSCs) are the therapeutic cells involved in the regenerative process [41]. Emerging evidence suggests that secretion of soluble factors could explain most of the beneficial effects of MSCs. They have multiple actions, including support of angiogenesis, modulation of inflammatory and immune reactions, protection against apoptosis, and stimulation of EPCs. MSCs have been shown to express and secrete factors essential for the process of angiogenesis, such as SDF-1, VEGF, basic fibroblast growth factor (FGF), or matrix metalloproteinases for the process of angiogenesis. MSCs are also able to stimulate endothelial cell migration and tube formation [42]. Moreover, MSCs have a vital role in stabilizing the new vasculature through their role as pericytes. These perivascular cells control proliferation and migration through interactions between endothelial cells [43].

Flow cytometric analysis of standard MSC markers revealed a significantly higher expression of CD44 and CD90 markers in CLI patients. The group of patients with NO-CLI had an excellent response to the application of bone marrow stem cells (BMCs) [43]. CD44 is a multistructural and multifunctional cell surface molecule involved in cell proliferation, cell differentiation, cell migration, and angiogenesis. Expanded BMCs enriched in CD90+ cells were efficient in the treatment of diabetic ulcers [44]. Moreover, MSCs play an essential role in the healing process via their immunomodulatory and anti-inflammatory properties, together with their antibacterial activity [45].

### 3.3. Indication/Contraindications of Cell Therapy in the Treatment of CLI

Cell therapy is currently indicated as an experimental treatment method in clinical trials and patients with severe PAD forms when standard treatment procedures have been exhausted without any possibility of further revascularization of the affected limb [46]. The contraindications include a presumed patient survival of fewer than six months, a known disease of the BM (e.g., lymphoma, leukemia, myelodysplastic syndrome, metastatic impairment of bone marrow), chronic renal insufficiency on dialysis therapy, or acute limb ischemia with a severe inflammatory reaction threatening the patient’s life implying the need of early amputation of the limb.

### 3.4. The Technique of Separation and Administration of Cell Therapy in the Treatment of CLI

Most clinical trials focusing on autologous cell therapy have used bone marrow mononuclear cells (BM-MNC). The hip bone’s top ridge is the most commonly used collection site; the collection itself is performed after a previous premedication and in local anesthesia. The procedure is performed under standard conditions with the administration of antibiotics. Another possibility is the separation of mononuclear cells from 60 mL of peripheral blood mononuclear cells (PB-MNC). Separation instruments must be used in suitable clinical setup, to obtain a cellular concentrate with the same biological function capable of supporting vascular growth in patients with PAD [47,48].

The protocols for autologous cell therapy use a broad spectrum of techniques for cell concentration. However, all used methods work similarly. They use a source material rich in pluripotent cells obtained from the patient’s BM (either from BM directly or after mobilizing them into the peripheral circulation). They concentrate mononuclear cells into a preparation suitable for injection application into the ischemic limb, (Figure 2a). After bone marrow aspiration (typically in a volume of 240 mL), the sample is treated with EDTA anticoagulation and subsequently separated for 15 min with gradient-density centrifugation, (Figure 2b). After centrifugation, the component rich in BM-MNC is aspirated. The isolate intended for PAD treatment is applied deep into the intramuscular space, along the presumed course of the affected limb’s crural arteries and into the surrounding area of the defect, (Figure 2c).

The intramuscular application under the control of ultrasound is theoretically unnecessary, considering the high migration activity of the isolate in tissues; nevertheless, given the distribution and easy accessibility in everyday practice, ultrasound control is recommended. After the procedure, the patient remains on bed rest for 24 h. Dressings are changed the following day, and the patient is discharged to home care. Standard changes in the defect dressings are proposed.

## 4. Results of Meta-Analyses

### Efficacy of Cell Therapy in the Treatment of CLI

After the promising results of preclinical studies suggesting a beneficial effect of BM-MNCs and MSCs on limb ischemia improvement, an increasing number of clinical trials emerged exploring the efficacy and safety of cell therapy in patients with CLI. Several prospective clinical trials performed between 2002 and 2016 studied the efficacy and safety of cell therapy in CLI treatment. The design of most published studies was of very high quality. Randomized prospective clinical trials as well as prospective controlled clinical trials (versus placebo or standard medical care) were performed. Results of the clinical trials were subsequently analyzed and thoroughly evaluated in several meta-analyses, which included more than 1500 patients [10,15,16,17,18,19,20,21,22,23,24].

An overview of meta-analyses with primary and secondary endpoints focused on the efficacy of the non-option CLI treatment is presented in Figure 3 and Table 2.

The primary aim of the presented meta-analyses was to assess the need to perform a major amputation of the limb (both below or above the knee) and compare the number of patients in whom complete healing of the defect was achieved. Conclusions of the included meta-analyses are similar. A statistically significant difference showing a positive effect of the cell therapy in CLI treatment was observed, together with a reduced number of major amputations of the limb and more healed defects in patients treated with cell therapy.

During the first year following the administration of cell therapy, Wang et al. reported eight times higher odds that the patients would not undergo limb amputation when compared with the control group (OR = 8.05, 95%CI (3.58; 18.08), *p* < 0.001). The odds of not undergoing limb amputation were approximately 22 times higher after 3 years (OR = 22.33, 95%CI (4.14; 120.50), *p* < 0.001) [18]. Liu Yumeng et al. reported three times higher odds of not undergoing limb amputation in the group with active treatment when compared with the control group (OR = 3.03, 95%CI (1.96; 4.55), *p* < 0.001) [19]. Liew et al. reported approximately two times higher odds of not undergoing limb amputation in the group of patients treated with cell therapy compared with the control group (OR = 1.85, 95%CI (1.15; 2.94), *p* = 0.010) [20]. All meta-analyses showed at least two times higher odds of complete defect healing compared to the control group. Liu Yumeng et al. demonstrated even a six times higher odds of defect healing in the treatment group [19].

Secondary aims assessed in the meta-analyses comprised values evaluating perfusion and oxygenation in the affected limb. The ankle-brachial pressure index (ABI), transcutaneous oxygen pressure (TcpO2), claudication interval, and pain manifestation in the limb were compared. The obtained results are presented in Table 2. The published data show an improved condition of the affected limb in patients undergoing cell therapy in all parameters. For example, Benoit et al. reported increased ABI values in 63.2% of patients in the reviewed studies included in their meta-analysis, improved TcpO2 in 76.9% of patients, pain reduction in almost 90%, and prolongation of the claudication interval in 89.5% of patients [17].

## 5. Discussion

Cell therapy represents a relatively safe therapeutic intervention, with a low risk of early complications in the course of and shortly after the procedure. The most frequent, although rarely observed, complication is bleeding from the collection site. The bleeding after bone marrow collection or peripheral venous access may be, nevertheless, quickly and effectively treated with compression. Benoit et al., in a group of more than 1200 patients, reported a formation of only one arteriovenous shunt after intramuscular BM administration into the limb. A spontaneous shunt occlusion was observed within 1 year after the procedure [17].

### 5.1. Bone Marrow Aspiration Concentrate (BMAC)

The incidence of anemia reported in the literature due to bone marrow collection for separation of the cellular concentrate is between 0.6% and 0.8%. [31] Considering the fact that the bone marrow aspirate comes from the patient’s own body, there is no risk of transferring infectious diseases (HIV, Hepatitis C, etc.) or adverse immunological reactions.

Potential risks associated with cell therapy administration include a progression of renal failure, diabetic retinopathy, cardiovascular risk, and possible cancer potentiation or acceleration. The cellular concentrate is applied deep into the intramuscular space, along the presumed course of the crural arteries. Thus, it is possible to anticipate that an intramuscular administration of BMC into the muscle tissue damaged with ischemia may lead to local rhabdomyolysis and worsening chronic renal insufficiency. However, the published studies have not confirmed any renal failure in relationship with cell therapy [17].

Endothelial progenitor cells participate in regenerative processes; however, they do not cause pathological vasculogenesis of retinal capillaries, which could worsen retinopathy [49]. No statistically significant difference in the incidence of cardiovascular conditions was reported in the above-presented meta-analyses.

Some tumors express chemotactic signals for the mobilization of monoclonal cells from the bone marrow; in the case of these cells, it is suspected that they may participate in the pathological vascularization of tumors [50,51]. Nevertheless, the mere presence of stem cells and a tumor is not sufficient for initiation of the process of pathological angiogenesis [52]. Wickersheim et al. have demonstrated in an animal model that endothelial progenitors derived from bone marrow are not present in the tumor endothelium of primary or secondary metastatic tumors [53].

No relationship between administration of cell therapy and an increased risk of cancer incidence has been demonstrated [50,51,52,53]. The risk of malignancy certainly increases with age and the presence of various physical, chemical, and biological risk factors. The incidence of PAD is frequently associated with smoking, which also represents a risk factor, significantly increasing the risk of malignant diseases. History of malignancy was considered a contraindication for cell therapy administration in most studies.

### 5.2. MSC-Derived Exosomes for Cell-Free Regenerative Therapy

A growing consensus exists that paracrine factors, including exosomes, mediate most of the stem cell therapy’s therapeutic effects. Exosomes are nanometer-sized membrane-bound vesicles, paracrine ingredients enclosed within lipid bilayers, and mediators of cell–cell communication. MSC-exosomes carry bioactive molecules, including growth factors, cytokines, or RNAs, which could be internalized by recipient cells to mediate intercellular communication. Therefore, the biological functions of MSC-exosomes are considered similar to MSCs, and MSC-derived exosome therapy is emerging as a promising strategy for the treatment of several diseases, including limb ischemia.

### 5.3. Adipose Tissue-Derived Stem Cells–Stromal Vascular Fraction (ASCs–SVF)

ASCs possess many phenotypic and functional similarities to bone marrow-derived MSCs (BM-MSCs). Moreover, unlike BM-MSCs, multipotent ASCs can be harvested at relatively high numbers (5 × 10^5^/g fat, a typical harvest of 500 g tissue) with minimally invasive techniques (i.e., lipoaspiration or lipectomy) from subcutaneous adipose tissue. ASCs rapidly proliferate in culture. This rapid expansion achieves cell quantities sufficient for treating large-volume lesions. Transplantation of ASCs is an emerging therapeutic option for addressing many intractable diseases, including cardiovascular diseases and peripheral artery disease (PAD). The nonadipocyte stromal vascular fraction (SVF) of human adipose tissue contains a population of mesenchymal stem cells (MSCs), intimately associated with blood vessels. Freshly isolated SVF and cultured cells displayed cell surface markers typical of human adipose tissue-derived populations [54].

Evidence suggests that ASC’s therapeutic effects are primarily mediated through paracrine mechanisms rather than through cellular transdifferentiation [55,56,57,58]. Secreted factors can be captured in a conditioned medium (ASC-CM) composed of a cocktail of beneficial growth factors and cytokines. Both individually and in combination, they demonstrate disease-modifying effects in animal models. The three predominant paracrine actions of adipose stem cell-secreted factors are trophic support of survival and repair of cells in diseased or injured tissues (VEGF, HGF), modulation of the immune system (IL-10, PGE2), and recruitment of endogenous stem and progenitor cells (GM-CSF, SDF-1). Moreover, these factors have the potential to work in concert to affect physiological improvements in diseased and injured tissues [59].

### 5.4. Regulatory Requirements

Most regulatory authorities have realized the need for early access to innovative therapies for unmet medical needs that may improve health care quality for life-threatening conditions. Since critical limb ischemia with its rising incidence is a significant health problem, accelerating innovative advanced therapies is essential.

Somatic cell therapies, including MSCs, are regulated as advanced therapy medicinal products (ATMPs). EU regulation allows member states to use the so-called “hospital exemption.” This exemption authorizes the hospital on the national level to use ATMPs without marketing authorization [Article 28. Regulation (EC) No 1394/2007 of the European Parliament and of the Council of 13 November 2007 on advanced therapy medicinal products and amending Directive 2001/83/EC and Regulation (EC) No 726/2004]. The intention and ratio of this clause are to allow noncommercial ATMPs with enough evidence for therapeutic use to be received by individual patients under the responsibility of the concrete hospital or medical practitioner.

In March 2016, the European Medicines Agency declared autologous bone marrow mononuclear cells (BM-MNC) as an advanced therapy medicinal product with the proposed indication to improve limb perfusion/restore blood flow. The cells are not intended to be used for the same essential function, but to regenerate and replace human tissue [European Medicines Agency—March 2016—Scientific recommendation on the classification of advanced therapy medicinal products].

Another important EU exemption is the “compassionate use” program for patients with life-threatening, long-lasting, or debilitating illness who cannot be treated by an authorized medical product. This program allows such a patient to access an investigational drug outside a clinical trial [Committee for Medicinal Products for Human Use (CHMP). Guideline on compassionate use of medicinal products, pursuant to article 83 of regulation (EC) No 726/2004. 2007; 2-3].

However, the balance between the progress of new technologies and innovations and scientific caution with emphasis on patients’ safety is necessary. Additionally, the homogenous approach towards regenerative therapy and innovation in the EU constitutes an appropriate barrier against stem cell tourism within Europe. Future studies should also focus on the possibility of stem cells application as adjunctive therapy in patients immediately after percutaneous revascularization of CLI limb and only one vessel below the knee run-off to intensify the healing process.

## 6. Conclusions

The aim of cell therapy in patients with CLI is to prolong limb survival, decrease the speed of symptom progression, and improve life quality. The available evidence-based medicine data show that this technique is safe, associated with minimum complications or adverse events, and relatively effective. The meta-analyses show that the reported odds ratio estimates of preventing amputation through cell therapy are mostly in the region 1.6–3 over the period of 12 months; more importantly, however, this number seems to steeply grow with a prolonged study period.

The majority of the adverse events were associated with hospitalization for complications related to disease progression, not to cell therapy, such as pain in the extremities and gastrointestinal disorders unrelated to cell therapy. In the considered RCTs, BM-derived cell therapy appeared to be relatively safe, and side effects were generally mild and transient.

Although stem cells extracted from bone marrow are at present the most investigated type, adipose tissue-derived stem cells–stromal vascular fraction (ASCs–SVF), an emerging method of cell therapy, is very promising due to its much easier extraction and, in addition, could be also used in diabetic patients or renal insufficiency whose bone marrow stem cells are of poor quality. However, the effectiveness of this method needs a lot of further research and validation. Cell therapy at present cannot be and is not an alternative to the already established techniques of PAD treatment. However, regenerative medicine may represent a future hope for the group of patients who have already exhausted the standard revascularization treatment and in whom symptomatic therapy does not bring the expected relief.

## Figures and Tables

**Figure 1 ijms-21-08999-f001:**
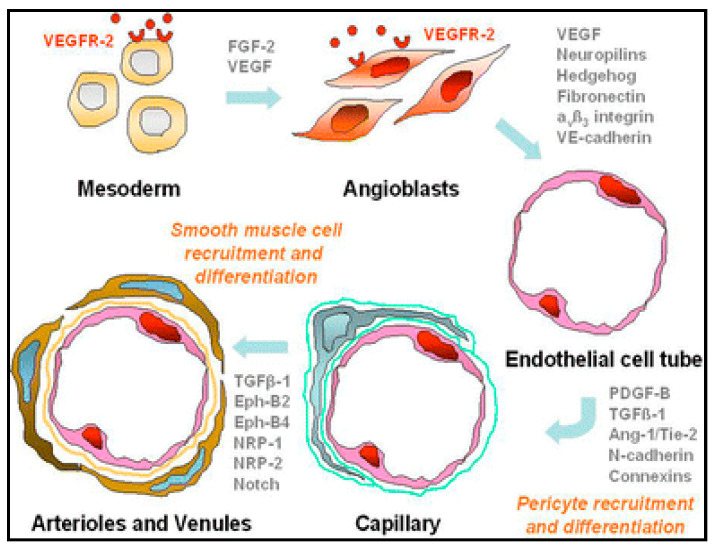
Scheme of vasculogenesis [13].

**Figure 2 ijms-21-08999-f002:**
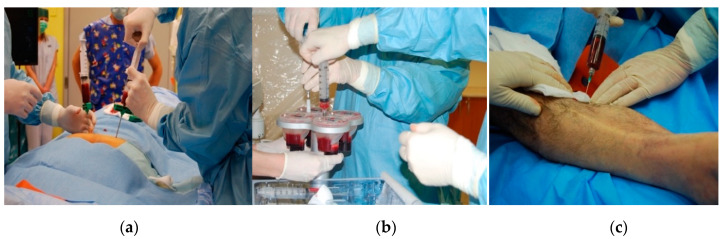
The technique of separation and administration of cell therapy: (**a**) harvesting BM-MNC (bone marrow mononuclear cells) from the hip bone. (**b**) Gradient—density centrifugation with bone marrow concentrate aspiration. (**c**) Intramuscular injection of the bone marrow concentrate along the calf vessels.

**Figure 3 ijms-21-08999-f003:**
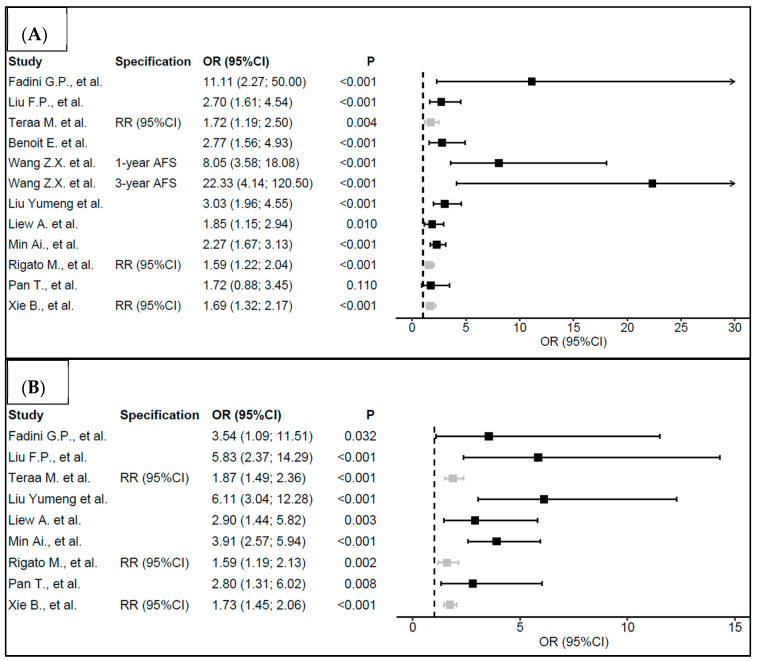
“Primary endpoints” evaluating the efficacy of non-option CLI treatment using the cell concentrate. The forest plots present (**A**) the odds ratios for nonamputation and (**B**) wound healing, respectively, for the group of patients treated with cell therapy. (OR = odds ratio, RR = risk ratio, 95%CI = 95% confidence interval, AFS = amputation-free survival.)

**Table 1 ijms-21-08999-t001:** Meta-analyses identified for further analysis.

Reference		Year of Publication	Records Screened	Records Excluded	Studies Assessed for Eligibility	RCT	Included Studies
Fadini G.P. et al.	[10]	2009	108	66	42	6	37
Liu F.P. et al.	[15]	2012	341	318	23	14	7
Teraa M. et al.	[16]	2013	2399	2385	14	12	12
Benoit E. et al.	[17]	2013	51	6	45	45	45
Wang Z.X. et al.	[18]	2014	102	27	75	31	9
Liu Yumeng et al.	[19]	2014	441	379	62	16	13
Liew A. et al.	[20]	2015	3910	3262	28	16	16
Min Ai. et al.	[21]	2016	526	468	58	25	25
Rigato M. et al.	[22]	2017	1532	1467	65	26	19
Pan T. et al.	[23]	2018	1495	1468	27	27	9
Xie B. et al.	[24]	2018	1130	662	23	23	23

**Table 2 ijms-21-08999-t002:** Secondary endpoints—in individual studies, primary endpoints were supplemented with secondary (minor) endpoints such as ABI, TcpO2, pain scale, and claudication interval. (MD = mean difference, SD = standard deviation, SE = standard error, SMD = standardized mean difference, OR = odds ratio, 95%CI = 95% confidence interval).

Study	Specification	ABI	TcpO2 (mmHg)	Pain (Scale 0–10)	Claudication Interval (m)
		**MD (95 %CI)**	***p***	**MD (95 %CI)**	***p***	**MD (95 %CI)**	***p***	**MD (95 %CI)**	*p*
Teraa M. et al.	Overall increase	0.12 (0.09; 0.15)	<0.001	14.28 (8.54; 20.02)	<0.001	−1.10 (−1.37; −0.83)	<0.001	178.73 (127.68; 229.78)	<0.001
Wang Z.X. et al.	Increase after 4–8 weeks	0.14 (0.07; 0.21)	<0.001	6.89 (6.17; 7.62)	<0.001	−0.01 (−1.44; 1.43)	0.990		
	Increase after 12 weeks	0.14 (0.00; 0.27)	0.050	1.95 (−7.41; 11.3)	0.680	−1.84 (−4.11; 0.44)	0.110		
	Increase after 24 weeks	0.14 (0.10; 0.19)	<0.001	20.35 (12.51; 28.19)	<0.001	−1.37 (−1.69; −1.04)	<0.001		
Rigato M., et al.	Overall increase	0.11 (0.07; 0.15)	<0.001	10.74 (4.93; 16.54)	<0.001	−0.74 (−1.12; −0.36)	<0.001	93.73 (−30.05; 217.51)	0.140
Xie B., et al.	Overall increase	0.13 (0.11; 0.15)	<0.001	12.22 (5.03; 19.41)	<0.001			144.84 (53.03; 236.66)	0.002
		**Mean ± SD or SE**		**Mean ± SD or SE**		**Mean ± SD or SE**		**Mean ± SD or SE**	
Fadini G.P., et al.	Before therapy	0.46 ± 0.04		22.8 ± 2.8		6.35 ± 0.43		75.7 ± 19.4	
	After therapy	0.63 ± 0.04	0.011	35.8 ± 2.9	<0.001	2.11 ± 0.37	< 0.001	402.3 ± 70.9	<0.001
		**SMD (95 %CI)**		**SMD (95 %CI)**		**SMD (95 %CI)**		**SMD (95 %CI)**	
Liu Yumeng et al.	Overall increase	0.65 (0.33; 0.97)	<0.001						
Min Ai., et al.	Overall increase	1.00 (0.63; 1.37)	<0.001	1.07 (0.39; 1.37)	0.002	−1.10 (−1.65; −0.56)	<0.001	1.12 (0.77; 1.47)	<0.001
		**Other**		**Other**		**Other**		**Other**	
Benoit E. et al.	No. studies with improvement/No. of all studies (in %)	24/38 (63.2)		20/26 (76.9)		33/37 (89.2)		17/19 (89.5)	
Liew A. et al.	OR (95 %CI) of improvement	5.91 (1.85; 18.86)	0.003

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
