# Peer review of "Current Status of Cell-Based Therapy in Patients with Critical Limb Ischemia"

_ijms, 2020, doi:10.3390/ijms21238999_

Round 1
Reviewer 1 Report
I have read with interest the manuscript “ Current Status of Cell-Based Therapy in Patients with Critical Limb Ischemia”
This is an interesting paper well written in terms of it following a logical progression and structure.
Thank you for submitting this article. I was pleased to receive it as a reviewer.
I have the following questions for the authors, which I believe, need to be addressed before publication:
DETAILED EVALUATION
Methods:
This section displays the main limitations of this study.
In this section the Authors should not report the techniques of separation and administration of cell therapy or indication/contraindications of cell therapy in the treatment of CLI, but Endpoints, Study selection, Data extraction, Statistical analysis.
The Authors should add the selection process of the selected papers in a Figure
The Authors should add a metanalysis with a Figure
Results
Authors should report the included trials and add a Table
Line 209: “…high amputation of the limb (amputation above the ankle joint)…”. Please re-edit this sentence. Amputations are classified in minor and major amputations. The major amputations are usually defined above or below the knee.
Author Response
Reviewer 1
I have read with interest the manuscript “Current Status of Cell-Based Therapy in Patients with Critical Limb Ischemia”. This is an interesting paper well written in terms of it following a logical progression and structure. Thank you for submitting this article. I was pleased to receive it as a reviewer.
Thank you for your kind and encouraging words.
I have the following questions for the authors, which I believe, need to be addressed before publication:
DETAILED EVALUATION
Methods:
This section displays the main limitations of this study. In this section the Authors should not report the techniques of separation and administration of cell therapy or indication/contraindications of cell therapy in the treatment of CLI, but Endpoints, Study selection, Data extraction, Statistical analysis.
Thank you for bringing this to our attention. We absolutely agree that the Methods section should deal with the methods used for preparation of this review rather than with methods used for cell therapy. We have amended the structure of the paper accordingly, changed the chapter name Methods to “Cell types and methods used in the cell therapy of CLI patients” and introduced the new chapter Methods, in which we describe the selection of papers.
However, please note that our paper is a narrative review focusing, in particular, on compiling information about the meta-analyses that have been performed so far. We have not performed a “meta-analysis of meta-analyses” ourselves (which would not yield meaningful results as many individual trials included in several meta-analyses would then be counted in several times). Rather, we have aggregated the information on individual meta-analyses and did our best to present their results in a concise manner. Hence, the structure you propose is not applicable to our paper.
The Authors should add the selection process of the selected papers in a Figure.
Similarly to the previous comment, our paper is a narrative review. Hence, although it would be possible to include the figure typical for meta-analyses or systematic reviews as suggested by you, it would be complicated by the fact that some studies were found not through search in databases but rather through cross-referencing of the used papers. Moreover, we have described the numbers in the new Methods section and such a figure would only duplicate the data already shown there.
The Authors should add a metanalysis with a Figure
As already explained in the comment to the first issue, performing a “meta-analysis of meta-analyses” would be scientifically incorrect and bias the results towards studies that were used in multiple meta-analyses.
However, to improve the readability of the comparisons of the meta-analyses, we included forest-plots for the primary endpoints of individual
Results
Authors should report the included trials and add a Table
Thank you for this suggestion. Table 1 listing the used meta-analyses was included.
Line 209: “…high amputation of the limb (amputation above the ankle joint)…”. Please re-edit this sentence. Amputations are classified in minor and major amputations. The major amputations are usually defined above or below the knee.
Thank you, the terminology was corrected as follows:
...the need to perform a major amputation of the limb (both below or above the knee)...
We would like to thank the Reviewer for his helpful comments that have definitely improved the quality of the manuscript.

Reviewer 2 Report
Dear Authors, this paper presents some interesting concerns, but this paper presents some criticisms I have raised.
The Author did not perform a review as they supposed on the title, they performed a book chapter focused on stem cells and CLI. The overall structure of the paper needed a substantial revision.
ABSTRACT
Aim: These are conclusions. Please correct
Method: You must include this session in the abstract and report your method for selection of papers.
Results: Please report more clear results.
INTRODUCTION
In this section, it is mandatory to report aims of the study, what the Authors want to add with this paper. The Authors discuss about CLI and possible alternative therapy, but it is difficult to understand the goal of this study. Another tools, it is difficult to understand that this paper represents a review.
Results of META-ANALYSES
This is the best section of the paper. You must report and describe it in abstract, methods, introduction and discussion.
Please report your personal data if available.
In this section, you must perform all possible statistical comparison between all papers you have selected.
This part need to be increased.
Please insert comment in discussion.
DISCUSSION
It is important to discuss what criteria have utilized to screen papers for this review.
Please comment different option of stem cells extrapolation and related results you have comprised in this review.
Please add comment about a possible role for stem cells as adjunct therapy and not only therapy for patients not suitable for revascularization.
CONCLUSION
Please report conclusion related to this review.
REFERENCES
Please add the infusion of vasoactive agents as a potential adjunct therapy:
De Caridi G, Massara M, Stilo F, Spinelli F, Grande R, Butrico L, de Franciscis S, Serra R. Effectiveness of prostaglandin E1 in patients with mixed arterial and venous ulcers of the lower limbs. Int Wound J. 2016 Oct;13(5):625-9. doi: 10.1111/iwj.12334. Epub 2014 Aug 5. PMID: 25091553.
Mirenda F, La Spada M, Baccellieri D, Stilo F, Benedetto F, Spinelli F. Iloprost infusion in diabetic patients with peripheral arterial occlusive disease and foot ulcers. Chir Ital 2005;6:731-735. PMID: 16400768
Author Response
Reviewer 2:
Dear Authors, this paper presents some interesting concerns, but this paper presents some criticisms I have raised.
The Author did not perform a review as they supposed on the title, they performed a book chapter focused on stem cells and CLI. The overall structure of the paper needed a substantial revision.
Thank you for this comment. It is true we have not performed a full systematic review but rather a narrative review, which has (as you aptly noted) rather a character of a book chapter. However, we aimed to prepare a paper with current data that would be suitable as a source of up-to-date information both for professionals only entering the field and for those who have already taken interest in this promising method but prefer concise summary on the current status of cell therapy from going through individual papers. In particular, we focused on the results of meta-analyses performed in this field and their comparison. The structure of the paper was also amended.
ABSTRACT
Aim: These are conclusions. Please correct
Thank you for pointing this out, we have amended the Aims in the Abstract as follows:
The aim of this narrative review was to provide an up-to-date critical overview of the knowledge and evidence-based medicine data on the position of cell therapy in the treatment of PAD. The current evidence on the cell-based therapy is summarized and future perspectives outlined, emphasizing the potential of exosomal cell-free approaches in patients with critical limb ischemia.
Method: You must include this session in the abstract and report your method for selection of papers.
Again, we thank you for this comment. The following was added into the Abstract:
Methods: Cochrane and PubMed databases were searched for keywords “critical limb ischemia cell therapy”. 589 papers were identified, 11 of which were reviews and 11 meta-analyses. These were used as the primary source of information, using cross-referencing for identification of additional papers
Besides, we have added a new section Methods into the paper, describing the search and study selection process.
Results: Please report more clear results.
We had discussed about your suggestion a lot as it definitely has merit. However, we have in the end concluded that as the Abstract is limited by the word count and we have already added words by adding the Methods section. In addition, as this is a review, which does not have results per se but rather compares results from numerous sources, we cannot present detailed results in the Abstract. Still, we added at least the most important numbers from the meta-analyses so the Results now read:
Meta-analyses focusing on cell therapy in PAD treatment confirm significantly greater odds of limb salvage in the first year after the cell therapy administration, the reported odds ratios of preventing amputation being mostly in the region 1.6 to 3. With prolonged observation times, however, these odds grew significantly. The odds of wound healing were at least two times higher when compared with the standard conservative therapy. Secondary endpoints of the available meta-analyses are also included in this review. Improvement of perfusion and oxygenation parameters in the affected limb, pain regression, and claudication interval prolongation are discussed.
INTRODUCTION
In this section, it is mandatory to report aims of the study, what the Authors want to add with this paper. The Authors discuss about CLI and possible alternative therapy, but it is difficult to understand the goal of this study. Another tools, it is difficult to understand that this paper represents a review.
We added the following paragraph to the end of the Introduction:
The use of cell therapy in treatment of critical limb ischemia is currently undergoing tumultuous development and although several reviews on this issue have been published, many of those are outdated by now. Also, several meta-analyses have been performed; none of the reviews, however, focused specifically on these meta-analyses and their detailed comparison. In this narrative review, we have collected the current information and aimed to provide an up-to-date critical overview of the knowledge and evidence-based medicine data on the position of cell therapy in the treatment of PAD. This review should serve as a source of information both to the professionals only entering this field and experienced experts looking for concise up-to-date information.
Results of META-ANALYSES
This is the best section of the paper. You must report and describe it in abstract, methods, introduction and discussion.
As detailed above, we emphasised this point in the other parts of our narrative review.
Please report your personal data if available. In this section, you must perform all possible statistical comparison between all papers you have selected. This part need to be increased. Please insert comment in discussion.
Including personal data in this paper would be most likely counterproductive for several reasons. The main reason is that they are already included in many of the meta-analyses and referenced in the text so their separate addition would not bring any new information. A statistical comparison of the studies including forest plots were added to this section.
DISCUSSION
It is important to discuss what criteria have utilized to screen papers for this review.
This information was added in the Methods section.
Please comment different option of stem cells extrapolation and related results you have comprised in this review.
We added the following comment into Conclusions:
Although stem cells extracted from bone marrow are at present the most investigated type, Adipose Tissue-Derived Stem Cells – Stromal Vascular Fraction (ASCs-SVF), an emerging method of cell therapy, is very promising due to its much easier extraction and, in addition, could be also used in diabetic patients or renal insufficiency whose bone marrow stem cells are of poor quality. However, the effectiveness of this method needs a lot of further research and validation.
Please add comment about a possible role for stem cells as adjunct therapy and not only therapy for patients not suitable for revascularization.
We added the following sentence to the end of the Discussion:
Future studies should also focus on the possibility of stem cells application as adjunctive therapy in patients immediately after or in combination with percutaneous revascularization of patients with CLI and only one vessel below the knee to intensify the healing process.
CONCLUSION
Please report conclusion related to this review.
Conclusions was extended by the aforementioned paragraph on various cell type extraction methods and by the following sentence:
The meta-analyses show that the reported odds ratios of preventing amputation through cell therapy are mostly in the region 1.6 to 3 over the period of 12 months; more importantly, however, this number steeply grows with a prolonged study period.
REFERENCES
Please add the infusion of vasoactive agents as a potential adjunct therapy:
We included the proposed therapy and references in the text. Thank you for the recommendation. In addition, we added additional references during editing the paper.
De Caridi G, Massara M, Stilo F, Spinelli F, Grande R, Butrico L, de Franciscis S, Serra R. Effectiveness of prostaglandin E1 in patients with mixed arterial and venous ulcers of the lower limbs. Int Wound J. 2016 Oct;13(5):625-9. doi: 10.1111/iwj.12334. Epub 2014 Aug 5. PMID: 25091553.
Mirenda F, La Spada M, Baccellieri D, Stilo F, Benedetto F, Spinelli F. Iloprost infusion in diabetic patients with peripheral arterial occlusive disease and foot ulcers. Chir Ital 2005;6:731-735. PMID: 16400768
- Min, Ai; Chang-Fu, Y.; Fu-Chun, X.; Shuang-Lu, Z.; Jian, H.; Cui-Ping, L.; Safety and efficacy of cell-based therapy on critical limb ischemia: A meta-analysis. Cytotherapy 2016, 18, 712-724.
- Rigato, M.; Monami, M.; Fadini, G.P.; Autologous Cell Therapy for Peripheral Arterial Disease. Systematic Review and Meta-Analysis of Randomized, Nonrandomized and Noncintrolled Studies. Circ.Res. 2017, 120, 1326-1340.
- Pan, T.; Wei, Z.; Fang, Y.; Dong, Z.; Fu, W.; Therapeutic efficacy of CD34+ cell-involved mononuclera cell therapy for no-option critical limb ischemia: A meta-analysis of randomized controlled clinical trials. VascularMedicine, 2018, 23, (3), 219-231.
- Xie, B.; Luo, H.; Zhang, Y.; Wang, Q.; Zhou, Ch.; Xu, D.; Autologous Stem Cell Therapy in Critical Limb Ischemia: A Meta-Analysis of Randomized Controlled trials. Stem Cells International, 2018, 7528464. doi: 10.1155/2018/7528464.eCollection 2018.
In all, we would like to thank the Reviewer for his apt and valuable comments that have, as we feel, improved the quality of the manuscript.
Round 2
Reviewer 1 Report
The authors successfully corrected the manuscript based on the reviewer's requests. In my opinion it can be accepted to be published.
Reviewer 2 Report
The Authors has been significantly improved this manuscript. Now it is good for publication on IJMS.